

**Understanding the Recent Trend of Haze Pollution in**
**Eastern China: Roles of Climate Change**
**H. J. Wang[1,2,3] and H. P. Chen[2,3,1]**
[1] {Collaborative Innovation Center on Forecast and Evaluation of Meteorological
Disasters, Nanjing University for Information Science and Technology, Nanjing,
China}
[2] {Nansen-Zhu International Research Center, Institute of Atmospheric Physics,
Chinese Academy of Sciences, Beijing, China}
[3] {Climate Change Research Center, Chinese Academy of Sciences, Beijing, China}
Correspondence to: H. J. Wang (wanghj@mail.iap.ac.cn)
**Abstract**
In this paper, the variation and trend of haze pollution in eastern China for winter of
1960-2012 were analyzed. With the overall increasing number of winter haze days in
the period, the five decades were divided into three sub-periods based on the changes
of winter haze days (WHD) in central North China ($30^{o}$N-$40^{o}$N) and eastern South
China (south of $30^{o}$N) for east of $109^{o}$E mainland China. Results show that WHD
kept gradual increasing during 1960-1979, overall stable during 1980-1999, and fast
increasing during 2000-2012. The author identified the major climate forcing factors
besides total energy consumption. Among all the possible climate factors, variability
of the autumn Arctic sea ice extent, local precipitation and surface wind during winter
is most influential to the haze pollution change. The joint effect of fast increase of
total energy consumption, rapid decline of Arctic sea ice extent and reduced
precipitation and surface winds intensified the haze pollution in central North China
after 2000. There is similar conclusion for haze pollution in eastern South China after
2000, with the precipitation effect being smaller and spatially inconsistent.





## 1. Introduction

In recent years, China has suffered from increased severe haze events that have caused strongly impacts on society, ecosystem, and human health. For example, the eastern China was hit by a prolonged and heavy haze event in January 2013, which made Beijing reaching its highest level of air pollution and led to the first haze orange alert in Beijing's meteorological history (e.g. Ding and Liu, 2014; Zhang et al., 2014). Furthermore, serious health problems have been induced from respiratory illness to heart disease, premature death, and cancer with the intensification of air pollution (Wang and Mauzerall, 2006; Xu et al., 2013; Xie et al., 2014). Thus, increased attentions have been reported to the issue of haze both from the government bodies and the general publics, and some air pollution prevention actions have been implemented and have stipulated strict controls on coal consumption, industry production, vehicles, etc.

Early studies have documented that the haze days are generally increase in economically developed eastern China but decrease in the less economically developed regions in China (e.g. Niu et al., 2010; Wu et al., 2010; Ding and Liu, 2014), and this increasing trend of haze is reported to be more pronounced since 2001 (Sun et al., 2013). Thus, the human activities, such as rapid urbanization and economic development, are generally considered as the major contributors to this long-term increasing trend of haze in eastern China (Wang et al., 2013). For example, in Beijing, vehicles are reported to be the biggest source of particulate matter 2.5 ($PM_{2.5}$), accounting for 25% of the pollution, and the coal combustion and cross-regional transport are the second greatest source, both accounting for 19%, despite some debates still exist (He et al., 2013; Zhang et al., 2013). Similar phenomenon can be observed in the other regions in China, such as in Chengdu city over southwestern China in which the secondary inorganic aerosols and coal combustion can account for $37 \pm 18\%$ and $20 \pm 12\%$ of the air pollutants, respectively (Tao et al., 2014).




Evidently, there is no doubt that there is a great role can be found from the human activities to the strongly increase of haze days in China. However, our deeply analysis in this study indicates that the variations of haze days show different trends in the past decades over eastern China, with increase in 1960-1979, no obvious change in 1980-1999 (even decrease over northern part of eastern China), and rapidly increase since 2000, which presents a disagreement with the persistently and rapidly increase of the total energy consumption over this region in the past. So, the impacts from the climate change must be considered when talking about the changes of haze events because the climate change can significantly influence the air pollution via variation of local atmospheric circulation. Some early studies have revealed that the increased haze days in eastern China may be associated with decreases of the surface wind speed (Xu et al., 2006; Gao et al., 2008; Niu et al., 2010) and the relative humidity in the atmosphere (Ding and Liu, 2014). Wu et al. (2008) indicate that the occurrences of heavy haze events in the Pearl River Delta region of China are generally concurrently with the stronger zonal circulations in the midtroposphere and weaker winds on the surface. Chen and Wang (2015) reveal that the severe haze events in boreal winter over northern China generally happen under a favorable atmospheric background, with the weakened northerly winds and the development of inversion anomalies in the lower troposphere, the weakened East Asian trough in the midtroposphere, and the northward East Asian jet in the high troposphere. Additionally, a recent study further reveals that the Arctic sea ice decline can intensify the haze pollution over eastern China and account for approximately 45-67% of the interannual to interdecadal variability of haze occurrences. However, the possible reasons for the different trends of haze days (varying from decades) over eastern China have not been revealed so far, although the ambient conditions of the haze occurrences have been well analyzed as well as the reason of its long-term increasing trend, which is thus to be our interest and topic in this study.

## 2. Data and Methods





The monthly haze day data for 756 meteorological stations in China during
1960-2013 have been collected by the National Meteorological Information Center of
the China Meteorological Administration. For the site-observation, it was rejected if
there are miss values in time series. Thus a subset of total 542 stations is selected. We
focus our analysis on haze pollution over eastern China (east of $109^o$E, south of $40^o$N,
mainland China) in this study. As have been indicated, more than 40% haze pollution
occurred in boreal winter (current year December and following year
January-February), hence we focus on the winter season. We here after focus our
analysis in two regions, R1 (east of $109^o$E in $30^o$N-$40^o$N) and R2 (east of $109^o$E, and
south of $30^o$N) in mainland China. Haze day is defined as the average in the region R1
or R2. The Arctic sea ice extent (ASI) is calculated from the Hadley Centre
(HadISST1) with $1^o \times 1^o$ resolution for 1870-2013 (Rayner et al, 2003). The autumn
ASI index is calculated as the area-averaged sea ice extent in the region of north $45^o$N.
The annual statistics of total energy consumption that providing for each province in
China are obtained from the journal of 'China Statistical Yearbook' that published
every year.
**3. Results**
Heavy haze events can not only strongly affect the traffic but also induce serious
health problems from respiratory illnesses to heart disease, premature death and
cancer (Pope and Docheru, 2006; Wang and Mauzerall, 2006). The intensified air
pollution in China can be more or less attributed to the increased emissions of
pollutants into the atmosphere as the result of rapid economic development thus fast
increase of fossil fuel energy consumption and urbanization. Meanwhile, climate
change can also significantly influence the air pollution via variation of local
atmospheric circulation and precipitation.
As indicated by numerous studies, air pollution has generally been intensified in
eastern China in past half-century, with more haze days and increased $PM_{2.5}$



concentration during winter and spring (e.g. Wang et al, 2015). However, based on
our current studies, recent trend during 2000-2012 is different from that during
1980-1999 or 1960-1979 (Fig. 1). During 1960-1979, there is a general consistent
increasing trend of winter haze days (WHD) in Beijing-Tianjin-Hebei area and in the
lower reaches of Yangtze River Valley. There is no significant trend southeastern
coastal region of China. In the second period (1980-1999), there are generally
increasing trends south of $30^{o}$N but some decreasing trends in regions between $30^{o}$N
and $40^{o}$N in eastern China. During recent period (2000-2012) there are generally large
increasing trends in the region south of $40^{o}$N in eastern China. During all the three
period, there is no significant trend in northeastern China and eastern Inner-Mongolia.
Thus, our first question is why there are some decreasing trends of WHD during the
second period (1980-1999) when the rapid economy has been growing continuously
from late 1970s up to present. We then plotted the WHD together with total energy
consumption in R1 and R2 (Fig. 2). We found that WHD keeps gradual increasing
during the first period, remains stable or slightly decreases during the second period,
and then increases fast along with the rapid increase of total energy consumption
during the recent period. Therefore, the contradiction between the no-increasing
WHD and increasing energy consumption during the second period must be explained
by other factors, most reasonably, some climate factors.
One of the possible major climate factors is the Arctic sea ice extent (Deser et al.,
2010; Liu et al., 2012; Li and Wang, 2013; Li and Wang, 2014), whose relationship
with the haze pollution in eastern China was first indicated by Wang et al. (2015).
Here we show the apparent out-of-phase interannual relationship between the Autumn
Arctic sea ice extent and WHD for both R1 and R2 in Fig. 3, with high correlation
coefficients of -0.70 and -0.87 respectively during 1960-2012, -0.60 and -0.82
respectively during 1980-2012. Meanwhile the WHDs in R1 and R2 are temporally
correlated each other at 0.75 during 1960-2012 in the interannual variability. With the
significant impact of sea ice extent on the haze pollution, the fact that sea ice extent
remains generally stable can largely explain the no-increase of WHD during the





second period even along with economic development and total energy consumption
increase. In addition, the rapid decline of the sea ice extent in recent two decades can
also largely explain the fast increase of WHD in both northern and southern areas of
eastern China.
Precipitation change is another important factor that has significant impact on the
haze pollution, via the wet removal effect of atmospheric pollutants. Here we plot the
spatial distribution of the linear trend of station winter precipitation in eastern China
for each of the three periods (Fig. 4). It is clear that R2 has generally increasing trend
of precipitation during the first and second periods while R1 has apparent decreasing
trend during the third period. Therefore, the precipitation trends favor WHD
decreasing in R2 in the first and second periods and favor WHD increasing in R1
during the third period. In this regard, the impacts of both the sea ice extent and
precipitation trends in R1 help to intensify the haze pollution in the central North
China (R1) in recent period. While the precipitation trend in R2 (R1) is generally
small in recent period (first two periods), thus has smaller impacts on WHD as
compared to sea ice extent.
The simultaneous WHD-precipitation correlation coefficient is -0.11 and -0.16
respectively for R1 and R2 during 1961-2011. However, the WHD-precipitation
correlation coefficient is -0.60 and -0.41 respectively for R1 and R2 during 1980-2011.
Besides, we should not neglect the effect of changing surface winds. As shown in Fig.
5, there is generally weak reduction of surface winds in eastern China before year
2000, but spatially inconsistent trends of surface wind after 2000. Region R2 has the
upward trend of surface wind after 2000, while R1 has upward and downward trends
respectively in the north and south parts of the region.
Therefore, the precipitation trends in eastern China and the sea ice extent can explain
larger proportion of WHD variance since 1980s in eastern China besides emission of
pollutants by human beings. After the year 2000, from climate change perspective, the
intensified WHD in R1 is a joint effect of sea ice decline and precipitation and surface



wind decrease whereas the intensified WHD in R2 is mainly induced by the sea ice
decline (the surface wind weak increase is not favorable to WHD increase).
Another widely concerned question is, has the governmental control on pollutant
emissions received positive effect? Based on our analysis, the answer is affirmative.
This can be demonstrated by comparing the PM2.5 content in large city like Beijing,
Tianjin, Hangzhou, Xi'an, Changchun, Shanghai, and Guangzhou between 2003 and
2013, where all the cities have much reduced PM2.5 content in 2013 than 2003 during
summer season (Cao et al., 2014). However, there has been no improvement of air
quality for winter season. Then, how to understand such difference between air
quality change between summer and winter? The key impact factor is the Artic sea ice
extent. On one hand, the winter atmospheric circulation in eastern China is
significantly modulated by the preceding autumn Arctic sea ice extent thus the sea ice
decline can intensify the haze pollution in eastern China even though the total
emission of pollutant into the atmosphere has been reduced. On the other hand, sea ice
extent has no significant influence on the summer atmospheric circulation, thus the
effect of cutting off the pollutant emission can be evidently observed. In other words,
the winter haze pollution would be more serious if the government has not controlled
the pollutant emission after the year 2000. Definitely, controls on the pollutant
emission always have positive effects and should be always encouraged.

**4. Conclusions and discussions**
Based on our above analysis, the Arctic sea ice extent has the most apparent impacts
on the haze pollution in eastern China among other climate factors including
precipitation and surface wind since 1980s. After the year 2000, the sea ice decline
and precipitation decrease in central North China jointly intensified the haze pollution,
whereas the net effects of sea ice decline and surface wind increase also intensified
the haze pollution in eastern South China. Our overall analysis and conclusions are
schematically summarized in Fig. 6.





However, two other points should be addressed here. The first point relates the inter-correlation among sea ice extent, precipitation, and surface winds. Based on our previous study (Wang et al., 2015), the Arctic sea ice decline may favor the Rossby wave activity weakening in eastern China south of $40^{o}$N thus leads to the precipitation decrease during winter season. Meanwhile, the change of sea ice extent may also have moderate impacts on both the zonal and meridional surface winds in eastern China. Secondly, recent trend after 2000 should be paid more attention. As we concluded above, both the Arctic sea ice decline and the precipitation decreasing in central North China, along with the total energy consumption increase, favors the haze pollution intensifying. In eastern South China, there are two apparent factors (sea ice decline and total energy consumption increasing) that help to intensifying the haze pollution except the precipitation. In addition, the surface wind keeps overall decreasing in central North China reflects the East Asian winter monsoon weakening after 1960 particularly after mid-1980s (Wang and He, 2012).

With the projected sea ice extent decrease (Kirtman et al., 2013), weakening of the winter East Asian monsoon wind (Wang et al., 2013) and total energy consumption increase, the haze pollution in eastern China may continue to be a serious problem in the near future. There are already series of governmental plans to address the air pollution issues in Beijing-Tianjin-Hebei area and Yangtze-River Delta as well as the Pearl River Delta even though the future climate change is not favorable to the air pollution reduction.

In Fig. 2, the year-to-year variation for summer haze days (SHD) is shown as well by the blue curve, indicating slight trend and rapid increase before and after 2000 for the two regions. Thus the intensification of the haze pollution in eastern China after 2000 is significant both in winter and summer. Changes of summer rainfall and near-surface wind should be directly associated with the SHD trend. Even though we did not find significant correlation between the SHD and the Arctic sea ice extent in the year-to-year variability, the SHD increase after 2000 may also be related to the Arctic sea ice. In addition, as shown in Zhu et al. (2011), the Pacific Decadal



Oscillation (PDO) phase change in late 1990s may have impacts on the summer
atmospheric circulation and precipitation changes in eastern China. Therefore
understanding of the climate mechanisms for the SHD change calls more
investigations from both local and remote perspectives.
**Acknowledgement**s
This research was supported by National Natural Science Foundation of China
(Grants 41421004 and 41130103).

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





**Figures**

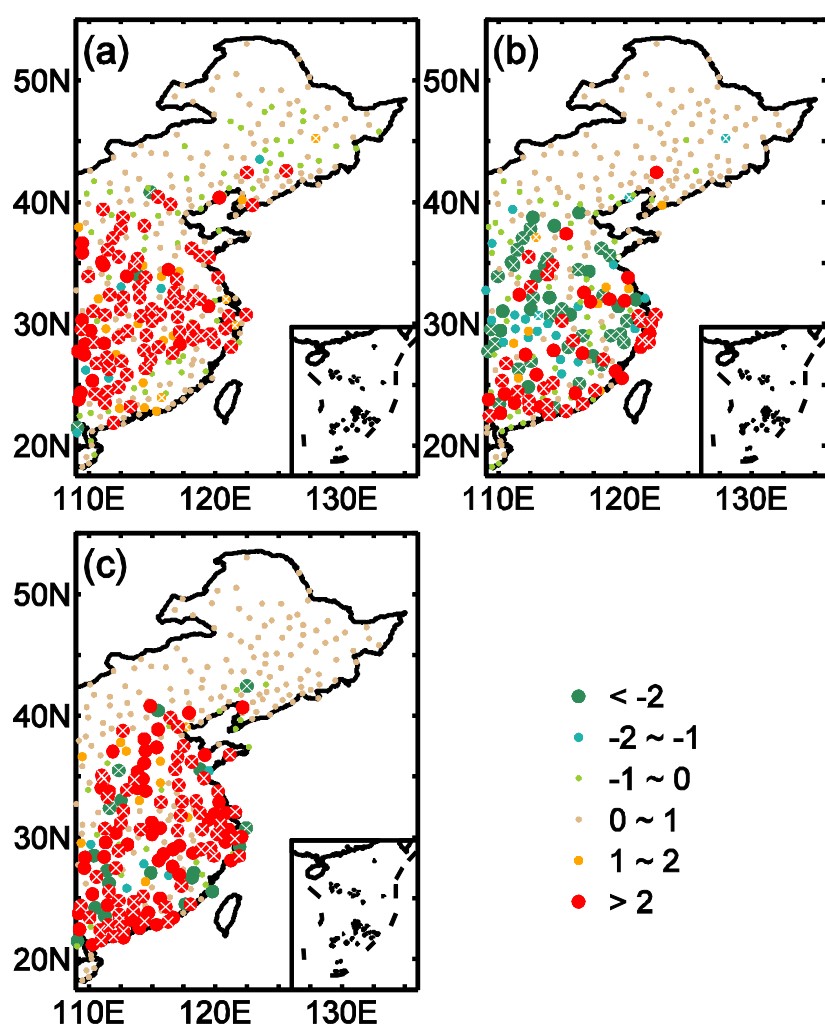

**Figure 1.** Linear trend of station winter haze days in the three periods: (a) 1960-1979,
(b) 1980-1999, and (c) 2000-2012. The circle with cross means the change is
significant at the 95% confidence level. Units: day·year$^{-1}$.





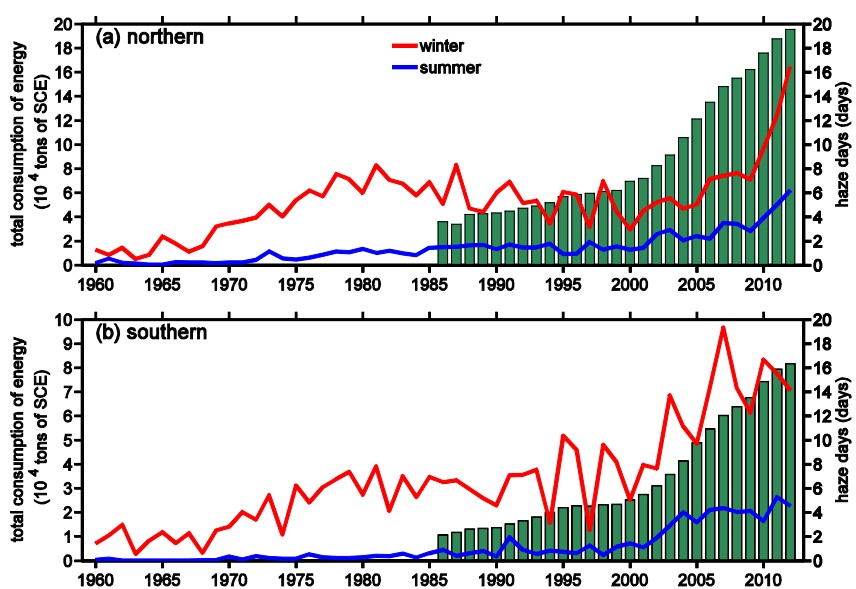

2   **Figure 2.** Time series for winter haze days (red curve), summer haze days (blue curve)

3   and total energy consumption (bar) for (a) region R1 (30$^o$N-40$^o$N) and (b) region R2

4   (south of 30$^o$N) in east of 109$^o$E of mainland China.





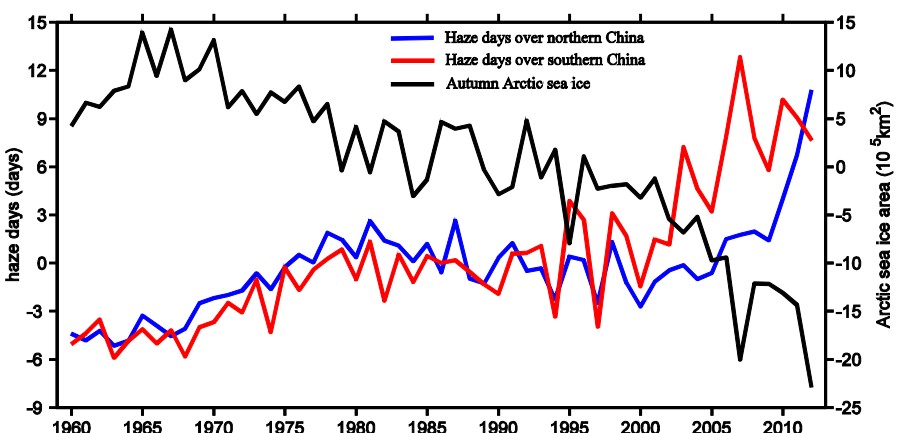

**Figure 3.** Temporal variations of winter haze days (WHD) for R1 (blue) and R2 (red), and autumn Arctic sea ice extent (ASI) (black). The results of correlation coefficient (CC) analysis are: CC(WHD-R1, WHD-R2)=0.75 in 1960-2012 and 0.58 in 1980-2012; CC(WHD-R1, ASI)= -0.70 in 1960-2012 and -0.60 in 1980-2012; CC(WHD-R2, ASI)= -0.87 in 1960-2012 and -0.82 in 1980-2012.





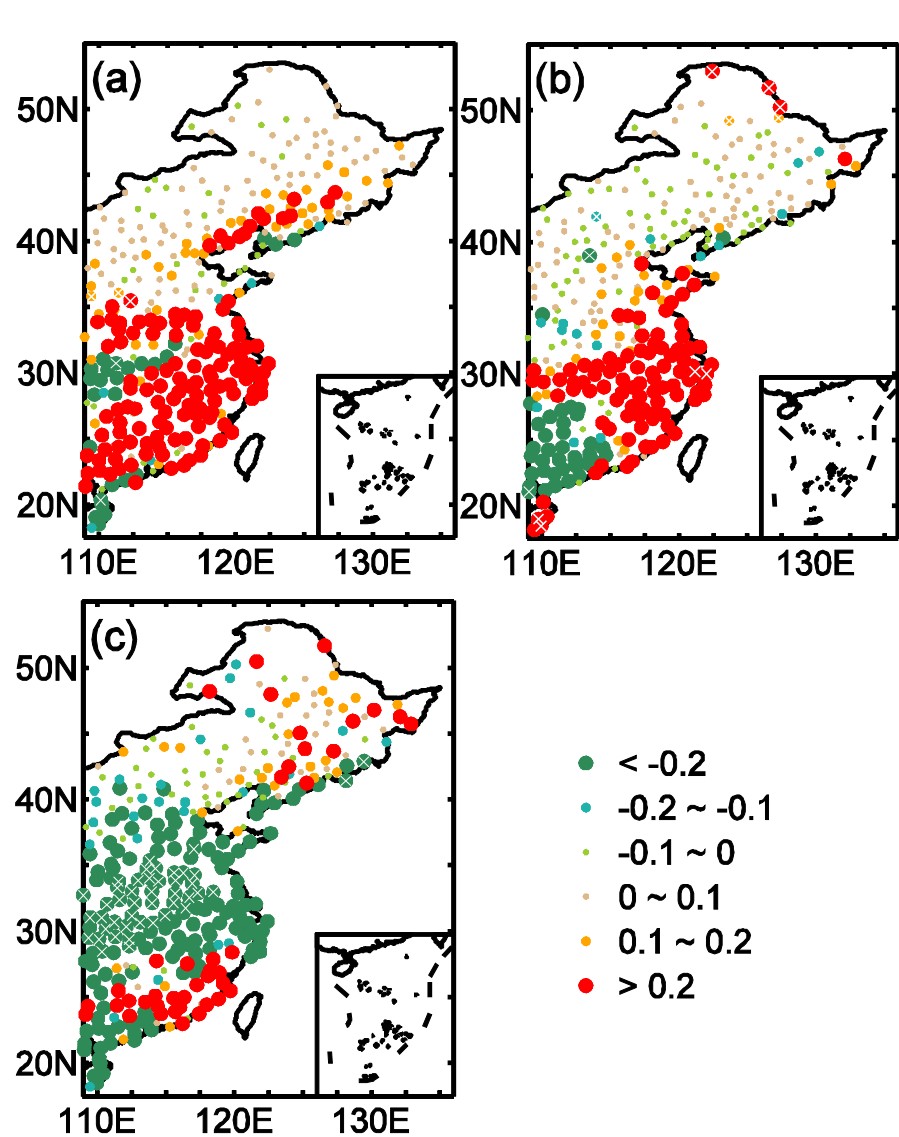

**Figure 4.** Linear trend of station winter precipitation (mm/day) in the three periods: (a) 1961-1979, (b) 1980-1999, and (c) 2000-2011. The circle with cross means the change is significant at the 95% confidence level.




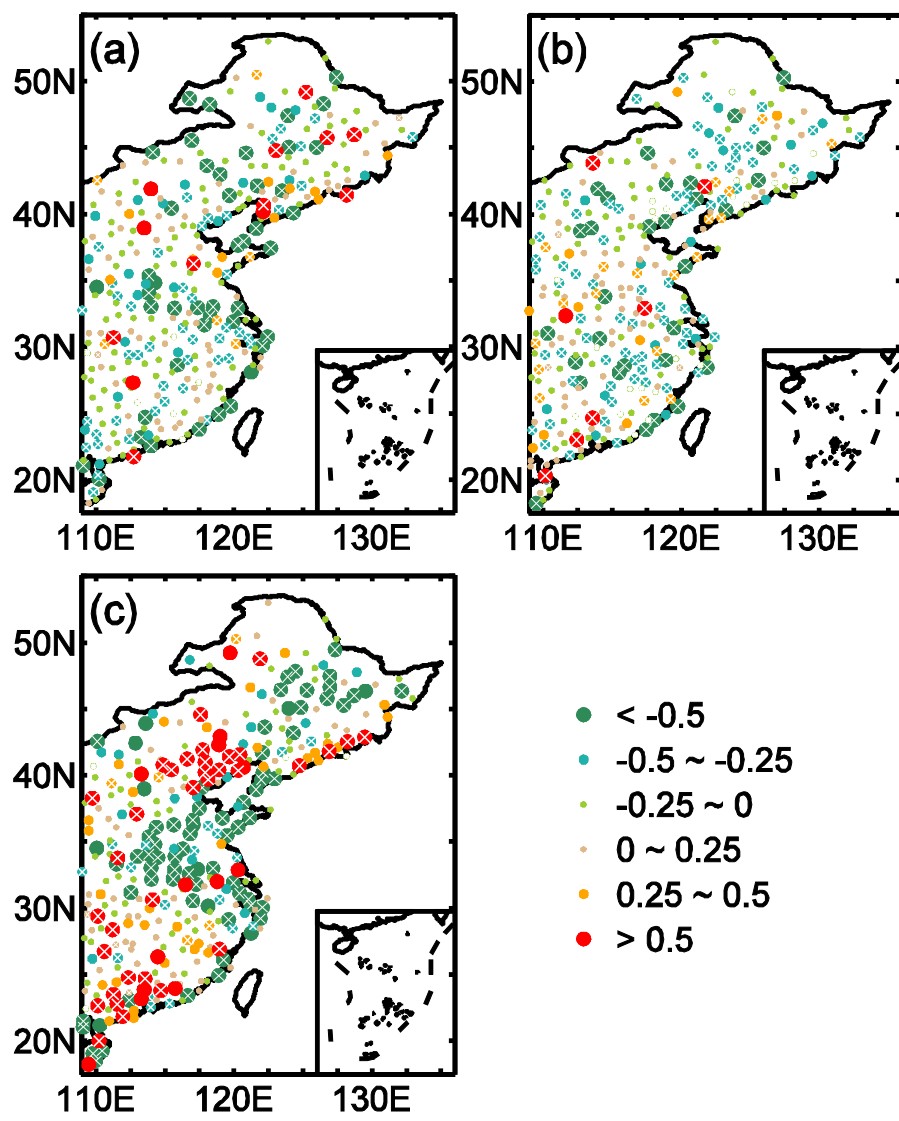

**Figure 5.** Same as in Figure 4 but for the winter surface wind speed (m/s).





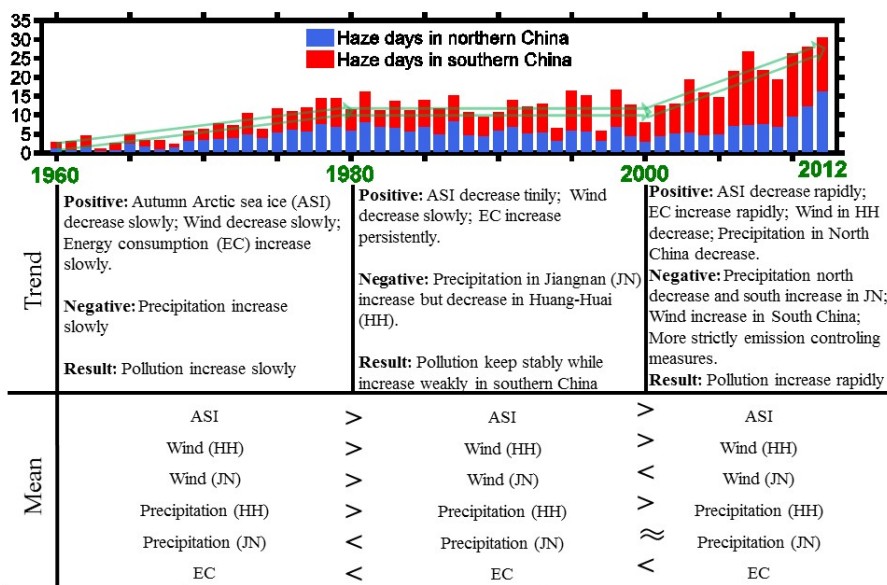

**Figure 6.** Summary of the haze pollution change in eastern China and various
influencing factors including climate change. The time series for winter haze days and
their linear trends are plotted on the top.

