# Peer review of "1. Introduction"

_Atmospheric Chemistry and Physics, 2015_

## Referee Comment (RC1) · Anonymous Referee #1 · 17 Feb 2016

In their contribution, the authors presented potential roles of climate variability and change in recent increased haze pollution events in east China. This paper could be a welcome reference in the literature.

In recent years, haze pollution has been a particularly acute issue in east China. The reasons behind recent increase of haze pollution events are complex. A number of studies have discussed this problem from human activity perspective, i.e., increased emissions into atmosphere due to urban and industrial pollution. However, little attention has been paid to this issue from climate variability and change perspective. The authors showed decadal trends in haze day in northeast and southeast China and their relationships with Arctic sea ice extent, precipitation and surface winds. The results can

improve our understanding of physical processes that influence haze variability in east China. Below I list some points, which the authors need to address.

1. Page 4/ Line9: Better outline the two regions (R1 and R2) in Figure 1. How many stations are used to calculate the averaged haze day in R1 and R2, respectively? Are the haze day trends sensitive to the number of stations used?

2. Page 4/Line 10: Explanation of the definition of haze day is needed.

3. Page 4/Line 13: "sea ice extent" should be "sea ice concentration".

4. Page 4/Line 14-16: Add "China Statistical Yearbook" in the reference list.

5. Page 5/Line 17: It seems that for R2, the variability of winter haze day becomes larger in recent years as compared to the first and second periods.

6. Page 6/Line 4: A little bit more discussion regarding the influence of Arctic sea ice loss on atmospheric circulation anomalies over east China is needed.

7. Page 6/Line 14-15: I suggest the authors add some discussion about the possible factors contributing to increased precipitation during the first and second period and decreased precipitation during the third period.

8. It is not clear how the authors define ">" and "<" signs in Figure 6? Relative to what? Please clarify.

9. Page 8/Line 15-21: I would like to suggest the authors to add more discussion about the projected changes in precipitation and surface winds over east China in near term based on recent studies, CMIP5 model projections.

---

## Author Comment (AC1) · 22 Feb 2016

Responses to Referee #1: In their contribution, the authors presented potential roles of climate variability and change in recent increased haze pollution events in east China. This paper could be a welcome reference in the literature. In recent years, haze pollution has been a particularly acute issue in east China. The reasons behind recent increase of haze pollution events are complex. A number of studies have discussed this problem from human activity perspective, i.e., increased emissions into atmosphere due to urban and industrial pollution. However, little attention has been paid to this issue from climate variability and change perspective. The authors showed decadal trends in haze day in northeast and southeast China and their relationships with Arctic

sea ice extent, precipitation and surface winds. The results can improve our understanding of physical processes that influence haze variability in east China. Below I list some points, which the authors need to address.

Reply: Thanks for your suggestions, which have been addressed point by point in the following and the corresponding corrections have been presented in the manuscript.

1. Page 4/ Line9: Better outline the two regions (R1 and R2) in Figure 1. How many stations are used to calculate the averaged haze day in R1 and R2, respectively? Are the haze day trends sensitive to the number of stations used?

Reply: (1) These two regions have been shown in Figure 1 [P14] as the following. According to the statistics, there are 112 stations included in R1 and 104 in R2 [P4 L11-12]. (2) As the suggested, further analysis is implemented and results indicate that the trends of haze days in these two regions show almost no sensitive to the number of stations used. However, a relative larger trend can be observed in southern China than that in northern China, which is also clear in Figure 1.

Figure 1. Linear trend of station winter haze days in the three periods: (a) 1960-1979, (b) 1980-1999, and (c) 2000-2012. R1 and R2 are the two regions that discussed in the text. The circle with cross means the change is significant at the 95% confidence level. Units: day•year-1.

2. Page 4/Line 10: Explanation of the definition of haze day is needed.

Reply: [P4 L3-5] The numbers of the monthly haze days that used in this study are directly derived from the collections by the National Meteorological Information Center of China Meteorological Administration. The haze days are generally determined according to the immediately weather phenomenon by the sophisticated observers. Thus, the definition of haze day has not been shown in this study. Additionally, the measured haze occurrences are also defined based on the observations of visibility and relative humidity according to specified criteria, which vary between organizations (e.g., World

Meteorological Organization and UK Met Office) and personal views (e.g., Vautard et al., 2009; Ding and Liu, 2014). In our early study, we adopted a comprehensive judgment method with visibility of less than 10 km, relative humidity at less than 90% and the wind speed lower than 7 m/s (Chen and Wang, 2015).

Ding, Y. H., and Liu, Y. J.: Analysis of long-term variations of fog and haze in China in recent 50 years and their relations with atmospheric humidity, Sci. China Earth Sci., 57, 36–46, 2014. Vautard, R., Yiou, P., and Oldenborgh, G.: Decline of fog, mist and haze in Europe over the past 30 years, Nat. Geosci., 2, 115–119, doi:10.1038/NGEO414, 2009. Chen, H. P., and Wang, H. J.: Haze Days in North China and the associated atmospheric circulations based on daily visibility data from 1960 to 2012, J. Geophys. Res. Atmos., 120, 5895–5909, doi:10.1002/2015JD023225, 2015.

3. Page 4/Line 13: "sea ice extent" should be "sea ice concentration".

Reply: [P4 L15-16] Sorry for misleading and this sentence has been reworded as "The autumn ASI index is calculated as the total sea ice extent in the region of Arctic".

4. Page 4/Line 14-16: Add "China Statistical Yearbook" in the reference list.

Reply: The 'China Statistical Yearbook' is published every year and there are at least 27 references from 1986 to 2012 which can be all found in the network. Thus, these references have not been listed in the manuscript.

5. Page 5/Line 17: It seems that for R2, the variability of winter haze day becomes larger in recent years as compared to the first and second periods.

Reply: It is actually true and it is apparent in Figure 2, which needs further studies in the future.

6. Page 6/Line 4: A little bit more discussion regarding the influence of Arctic sea ice loss on atmospheric circulation anomalies over east China is needed.

Reply: [P6 L6-11] Thanks for the suggestions. Actually, there are already some discussions can be found in section 4 for this perspective. As the suggestion, more discussion regarding this has been added in the current version of this study as "Early studies (e.g., Wang et al., 2015) have indicated that the reduction of autumn ASI can lead to positive sea level pressure anomalies in mid-latitude Eurasia, northward shift of track of cyclone activity in China and weak Rossby wave activity in eastern China during winter season. These atmospheric circulation changes favor less cyclone activity and more stable atmosphere in eastern China, resulting in more haze days there."

Wang, H. J., Chen, H. P., and Liu, J.: Arctic sea ice decline intensified haze pollution in eastern China. Atmos. Oceanic Sci. Lett., 8, 1-9, 2015.

7. Page 6/Line 14-15: I suggest the authors add some discussion about the possible factors contributing to increased precipitation during the first and second period and decreased precipitation during the third period.

Reply: Thanks for the suggestions. However, this article is mainly focused on the discussion about the influences of precipitation on the haze pollution, not the reasons for the precipitation changes. This suggestion as you proposed is beyond the scope of this article, thus the discussions about the precipitation changes have not been added in the current version.

8. It is not clear how the authors define ">" and "<" signs in Figure 6? Relative to what? Please clarify.

Reply: [P19] (1) ">" means "larger than" and "<" means "less than" in Figure 6, which have been clarified in the figure caption. (2) The comparisons are implemented among these three periods that discussed in the paper, i.e. the second period is compared with the first period and the third period is compared with the second period.

9. Page 8/Line 15-21: I would like to suggest the authors to add more discussion about the projected changes in precipitation and surface winds over east China in near term based on recent studies, CMIP5 model projections.

Reply: [P8 L22-27] Some discussions have been added as "Projections from CMIP5 models indicate that the low-level atmosphere tends to be more unstable and the atmosphere humidity will decrease in eastern China (Wang et al., 2015). Simultaneously, the winter precipitation in eastern China is projected to increase (Tian et al., 2015), but the surface winds decrease (Jiang et al., 2013). Thus there will be both favorable and unfavorable factors for haze occurrences in the near future based on the model projections. However, there is no doubt that, with the projected sea ice extent decrease (Kirtman et al., 2013), weakening of the winter East Asian monsoon wind (Wang et al., 2013) and total energy consumption increase, the haze pollution in eastern China may continue to be a serious problem in the near future. "

Wang, H. J., Chen, H. P., and Liu, J.: Arctic sea ice decline intensified haze pollution in eastern China. Atmos. Oceanic Sci. Lett., 8, 1-9, 2015. Tian, D., Guo, Y., and Dong, W. J.: Future changes and uncertainties in temperature and precipitation over China based on CMIP5 models. Adv. Atmos. Sci., 32(4), 487-496, doi:10.1007/s00376-014-4102-7, 2015. Jiang, Y., Luo, Y., and Zhao, Z. C.: Maximum wind speed changes over China. Acta Meteor. Sinica, 27(1), 63-74, doi:10.1007/s13351-013-0107-x, 2013. Kirtman, B. et al.: Near-term Climate Change: Projections and Predictability. In: Climate Change 2013: The Physical Science Basis. Contribution of Working Group I to the Fifth Assessment Report of the Intergovernmental Panel on Climate Change [Stocker, T. F., Qin, D., Plattner, G.-K., Tignor, M., Allen, S. K., Boschung, J., Nauels, A., Xia, Y., Bex, V., and Midgley, P. M. (eds.)]. Cambridge University Press, Cambridge, United Kingdom and New York, NY, USA, pp. 953–1028, doi:10.1017/CBO9781107415324.023, 2013. Wang H. J., He, S. P., and Liu, J. P.: Present and Future Relationship between the East Asian winter monsoon and ENSO: Results of CMIP5. J. Geophys. Res. Ocean, 118, 1-16, DOI:10.1002/jgrc.20332, 2013.

Please also note the supplement to this comment:
http://www.atmos-chem-phys-discuss.net/acp-2015-1009/acp-2015-1009-AC1-supplement.pdf

[Figure]

[Figure]

**Fig. 1.** Linear trend of station winter haze days in the three periods: (a) 1960-1979, (b) 1980-1999, and (c) 2000-2012. R1 and R2 are the two regions that discussed in the text.

**Supplement:**

**Responses to Referee #1:**

In their contribution, the authors presented potential roles of climate variability and change in recent increased haze pollution events in east China. This paper could be a welcome reference in the literature.

In recent years, haze pollution has been a particularly acute issue in east China. The reasons behind recent increase of haze pollution events are complex. A number of studies have discussed this problem from human activity perspective, i.e., increased emissions into atmosphere due to urban and industrial pollution. However, little attention has been paid to this issue from climate variability and change perspective. The authors showed decadal trends in haze day in northeast and southeast China and their relationships with Arctic sea ice extent, precipitation and surface winds. The results can improve our understanding of physical processes that influence haze variability in east China. Below I list some points, which the authors need to address.

**Reply:** Thanks for your suggestions, which have been addressed point by point in the following and the corresponding corrections have been presented in the manuscript.

1.  Page 4/ Line9: Better outline the two regions (R1 and R2) in Figure 1. How many stations are used to calculate the averaged haze day in R1 and R2, respectively? Are the haze day trends sensitive to the number of stations used?

**Reply:** (1) These two regions have been shown in Figure 1 **[P14]** as the following. According to the statistics, there are 112 stations included in R1 and 104 in R2 **[P4 L11-12]**.

(2) As the suggested, further analysis is implemented and results indicate that the trends of haze days in these two regions show almost no sensitive to the number of stations used. However, a relative larger trend can be observed in southern China than that in northern China, which is also clear in Figure 1.

[Figure]

**Figure 1.** Linear trend of station winter haze days in the three periods: (a) 1960-1979,

(b) 1980-1999, and (c) 2000-2012. R1 and R2 are the two regions that discussed in

the text. The circle with cross means the change is significant at the 95% confidence

level. Units: day·year[-1].

2.  Page 4/Line 10: Explanation of the definition of haze day is needed.

**Reply: [P4 L3-5]** The numbers of the monthly haze days that used in this study are

directly derived from the collections by the National Meteorological Information

Center of China Meteorological Administration. The haze days are generally

determined according to the immediately weather phenomenon by the sophisticated

observers. Thus, the definition of haze day has not been shown in this study.

Additionally, the measured haze occurrences are also defined based on the

observations of visibility and relative humidity according to specified criteria, which

vary between organizations (e.g., World Meteorological Organization and UK Met

Office) and personal views (e.g., Vautard et al., 2009; Ding and Liu, 2014). In our

early study, we adopted a comprehensive judgment method with visibility of less than

10 km, relative humidity at less than 90% and the wind speed lower than 7 m/s (Chen and Wang, 2015).

Ding, Y. H., and Liu, Y. J.: Analysis of long-term variations of fog and haze in China in recent 50 years and their relations with atmospheric humidity, Sci. China Earth Sci., 57, 36–46, 2014.

Vautard, R., Yiou, P., and Oldenborgh, G.: Decline of fog, mist and haze in Europe over the past 30 years, Nat. Geosci., 2, 115–119, doi:10.1038/NGEO414, 2009.

Chen, H. P., and Wang, H. J.: Haze Days in North China and the associated atmospheric circulations based on daily visibility data from 1960 to 2012, J. Geophys. Res. Atmos., 120, 5895–5909, doi:10.1002/2015JD023225, 2015.

3. Page 4/Line 13: "sea ice extent" should be "sea ice concentration".

**Reply: [P4 L15-16]** Sorry for misleading and this sentence has been reworded as "The autumn ASI index is calculated as the total sea ice extent in the region of Arctic".

4. Page 4/Line 14-16: Add "China Statistical Yearbook" in the reference list.

**Reply:** The 'China Statistical Yearbook' is published every year and there are at least 27 references from 1986 to 2012 which can be all found in the network. Thus, these references have not been listed in the manuscript.

5. Page 5/Line 17: It seems that for R2, the variability of winter haze day becomes larger in recent years as compared to the first and second periods.

**Reply:** It is actually true and it is apparent in Figure 2, which needs further studies in the future.

6. Page 6/Line 4: A little bit more discussion regarding the influence of Arctic sea ice loss on atmospheric circulation anomalies over east China is needed.

**Reply: [P6 L6-11]** Thanks for the suggestions. Actually, there are already some

discussions can be found in section 4 for this perspective. As the suggestion, more discussion regarding this has been added in the current version of this study as "Early studies (e.g., Wang et al., 2015) have indicated that the reduction of autumn ASI can lead to positive sea level pressure anomalies in mid-latitude Eurasia, northward shift of track of cyclone activity in China and weak Rossby wave activity in eastern China during winter season. These atmospheric circulation changes favor less cyclone activity and more stable atmosphere in eastern China, resulting in more haze days there."

Wang, H. J., Chen, H. P., and Liu, J.: Arctic sea ice decline intensified haze pollution in eastern China. Atmos. Oceanic Sci. Lett., 8, 1-9, 2015.

7. Page 6/Line 14-15: I suggest the authors add some discussion about the possible factors contributing to increased precipitation during the first and second period and decreased precipitation during the third period.

**Reply:** Thanks for the suggestions. However, this article is mainly focused on the discussion about the influences of precipitation on the haze pollution, not the reasons for the precipitation changes. This suggestion as you proposed is beyond the scope of this article, thus the discussions about the precipitation changes have not been added in the current version.

8. It is not clear how the authors define ">" and "<" signs in Figure 6? Relative to what? Please clarify.

**Reply: [P19]** (1) ">" means "larger than" and "<" means "less than" in Figure 6, which have been clarified in the figure caption.

(2) The comparisons are implemented among these three periods that discussed in the paper, i.e. the second period is compared with the first period and the third period is compared with the second period.

9. Page 8/Line 15-21: I would like to suggest the authors to add more discussion

about the projected changes in precipitation and surface winds over east China in near term based on recent studies, CMIP5 model projections.

[revised manuscript text omitted]

---

## Referee Comment (RC2) · Anonymous Referee #2 · 1 Mar 2016

The impact of climate change on air quality is an important cross-disciplinary topic. Previous studies were mainly conducted by using numerical models and there is limited number of studies based on analysis of observational data, especially in China. This study presented a very interested analysis on the trend of number of haze days and investigated the influences from Artic sea ice extent, precipitation and surface wind speed for different decades based on measurements at 756 ground station during 1960-2012. This paper is generally well-written and provides a different but unique angle of view from climatologist on the trend of air pollution in eastern China. I would like to recommend its publication in Atmospheric Chemistry and Physics after the following minor/technical points addressed appropriately.

[Figure]

Main comments:

1)If I understand it correctly, the "haze day data" (i.e. the term defined as "monthly haze day data" at L1, Page 4.) used in the trend analysis is the "total number of haze day in a month". In the text, this term is not very clear defined. It will be better to give some details of the data in Sect. 2 and clarified this term because it is not widely used in countries other than China. Also, it will be better to include some previous works e.g. Chen et al., 2015, which used the same dataset, as a reference in this section.

2)The discussions in the last paragraph of Sect. 3 (L3-L19, Page 7): The authors tried to discuss the reasons of the contradict trends between haze pollution and emission control. One point worthwhile to be mentioned here is that the trend of haze pollution based on "haze day data" actually is the trend of frequency (of haze day) but not the averaged pollution concentrations. The former might link more with the change in occurrence frequency of extremely stagnant weather, which was influenced by natural climate variability (Zhang et al., 2016), but the latter will be more related to the emission and control measures. Here a comparison with the two variables should consider the differences, at least mention the possible influence. In additional, this part probably can be moved into the Section 4 as the last discussion point highlighted for policy makers.

Reference: Zhang, Y. et al., Impact of synoptic weather patterns and inter-decadal climate variability on air quality in the North China Plain during 1980-2013, Atmos. Environ.,124, 119-128, 2016.

Minor comments:

1)Line 21-22, page 2: "particulate matter 2.5", Please change it to "fine particulate matter (PM2.5)" or "particulate matter with mean aerodynanmic diameter less than 2.5 micrometers".

2)Line 20-21, Page 3: "a recent study further reveals that", should you have a reference here? For example, "a recent study by Wang (2015) further revealed that".

3)Figure 6: What does "Jiangnan(JN)" in the figure notes mean? Maybe change it to "South China".

---

## Author Comment (AC2) · 4 Mar 2016

Responses to Referee #2: The impact of climate change on air quality is an important cross-disciplinary topic. Previous studies were mainly conducted by using numerical models and there is limited number of studies based on analysis of observational data, especially in China. This study presented a very interested analysis on the trend of number of haze days and investigated the influences from Artic sea ice extent, precipitation and surface wind speed for different decades based on measurements at 756 ground station during 1960-2012. This paper is generally well-written and provides a different but unique angle of view from climatologist on the trend of air pollution in eastern China. I would like to recommend its publication in Atmospheric Chemistry

and Physics after the following minor/technical points addressed appropriately. Reply: Thanks for your suggestions, which have been addressed point by point in the following and the corresponding corrections have been presented in the manuscript.

Main comments: 1) If I understand it correctly, the "haze day data" (i.e. the term defined as "monthly haze day data" at L1, Page 4.) used in the trend analysis is the "total number of haze day in a month". In the text, this term is not very clear defined. It will be better to give some details of the data in Sect. 2 and clarified this term because it is not widely used in countries other than China. Also, it will be better to include some previous works e.g. Chen et al., 2015, which used the same dataset, as a reference in this section. Reply: (1) Yes, the "monthly haze day" referred in this study actually is the "total number of haze in a month", which has been clarified in the paper. (2) The previous works (e.g. Chen et al., 2015) as you referred is the paper by Chen and Wang, 2015, JGR? If yes, we should clarify that the haze day in that work is defined using visibility, relative humidity, and wind speed, which is different from the data that used in this study. However, similar results can be found between these two datasets. The haze days that used in this study are generally determined according to the immediately weather phenomenon by the sophisticated observers, which have been used in another work (Wang et al., 2015). These corrections have been clarified in the current version of the manuscript.

2) The discussions in the last paragraph of Sect. 3 (L3-L19, Page 7): The authors tried to discuss the reasons of the contradict trends between haze pollution and emission control. One point worthwhile to be mentioned here is that the trend of haze pollution based on "haze day data" actually is the trend of frequency (of haze day) but not the averaged pollution concentrations. The former might link more with the change in occurrence frequency of extremely stagnant weather, which was influenced by natural climate variability (Zhang et al., 2016), but the latter will be more related to the emission and control measures. Here a comparison with the two variables should consider the differences, at least mention the possible influence. In additional, this part probably can

be moved into the Section 4 as the last discussion point highlighted for policy makers. Reference: Zhang, Y. et al., Impact of synoptic weather patterns and inter-decadal climate variability on air quality in the North China Plain during 1980-2013, Atmos. Environ.,124, 119-128, 2016. Reply: As your suggestion, more discussions have been added and this part has been also moved to Section 4.

Minor comments: 1) Line 21-22, page 2: "particulate matter 2.5", Please change it to "fine particulate matter (PM2.5)" or "particulate matter with mean aerodynanmic diameter less than 2.5 micrometers". Reply: Corrected. 2) Line 20-21, Page 3: "a recent study further reveals that", should you have a reference here? For example, "a recent study by Wang (2015) further revealed that". Reply: Corrected. 3) Figure 6: What does "Jiangnan (JN)" in the figure notes mean? Maybe change it to "South China". Reply: Corrected. "Jiangnan (JN)" has been changed to R2 and "Huang-Huai (HH)" to R1, which have been labeled in Figure 1.

Please also note the supplement to this comment:
http://www.atmos-chem-phys-discuss.net/acp-2015-1009/acp-2015-1009-AC2-supplement.pdf

**Supplement:**

**Responses to Referee #1:**

In their contribution, the authors presented potential roles of climate variability and change in recent increased haze pollution events in east China. This paper could be a welcome reference in the literature.

In recent years, haze pollution has been a particularly acute issue in east China. The reasons behind recent increase of haze pollution events are complex. A number of studies have discussed this problem from human activity perspective, i.e., increased emissions into atmosphere due to urban and industrial pollution. However, little attention has been paid to this issue from climate variability and change perspective. The authors showed decadal trends in haze day in northeast and southeast China and their relationships with Arctic sea ice extent, precipitation and surface winds. The results can improve our understanding of physical processes that influence haze variability in east China. Below I list some points, which the authors need to address.

**Reply:** Thanks for your suggestions, which have been addressed point by point in the following and the corresponding corrections have been presented in the manuscript.

1. Page 4/ Line9: Better outline the two regions (R1 and R2) in Figure 1. How many stations are used to calculate the averaged haze day in R1 and R2, respectively? Are the haze day trends sensitive to the number of stations used?

**Reply:** (1) These two regions have been shown in Figure 1 **[P14]** as the following. According to the statistics, there are 112 stations included in R1 and 104 in R2 **[P4 L11-12]**.

(2) As the suggested, further analysis is implemented and results indicate that the trends of haze days in these two regions show almost no sensitive to the number of stations used. However, a relative larger trend can be observed in southern China than that in northern China, which is also clear in Figure 1.

[Figure]

2  **Figure 1.** Linear trend of station winter haze days in the three periods: (a) 1960-1979,

3  (b) 1980-1999, and (c) 2000-2012. R1 and R2 are the two regions that discussed in

4  the text. The circle with cross means the change is significant at the 95% confidence

5  level. Units: day year$^{-1}$.

7  2.  Page 4/Line 10: Explanation of the definition of haze day is needed.

8  **Reply: [P4 L3-5]** The numbers of the monthly haze days that used in this study are

9  directly derived from the collections by the National Meteorological Information

10  Center of China Meteorological Administration. The haze days are generally

11  determined according to the immediately weather phenomenon by the sophisticated

12  observers. Thus, the definition of haze day has not been shown in this study.

13  Additionally, the measured haze occurrences are also defined based on the

14  observations of visibility and relative humidity according to specified criteria, which

15  vary between organizations (e.g., World Meteorological Organization and UK Met

16  Office) and personal views (e.g., Vautard et al., 2009; Ding and Liu, 2014). In our

17  early study, we adopted a comprehensive judgment method with visibility of less than

10 km, relative humidity at less than 90% and the wind speed lower than 7 m/s (Chen and Wang, 2015).

Ding, Y. H., and Liu, Y. J.: Analysis of long-term variations of fog and haze in China in recent 50 years and their relations with atmospheric humidity, Sci. China Earth Sci., 57, 36–46, 2014.

Vautard, R., Yiou, P., and Oldenborgh, G.: Decline of fog, mist and haze in Europe over the past 30 years, Nat. Geosci., 2, 115–119, doi:10.1038/NGEO414, 2009.

Chen, H. P., and Wang, H. J.: Haze Days in North China and the associated atmospheric circulations based on daily visibility data from 1960 to 2012, J. Geophys. Res. Atmos., 120, 5895–5909, doi:10.1002/2015JD023225, 2015.

3. Page 4/Line 13: "sea ice extent" should be "sea ice concentration".

**Reply: [P4 L15-16]** Sorry for misleading and this sentence has been reworded as "The autumn ASI index is calculated as the total sea ice extent in the region of Arctic".

4. Page 4/Line 14-16: Add "China Statistical Yearbook" in the reference list.

**Reply:** The 'China Statistical Yearbook' is published every year and there are at least 27 references from 1986 to 2012 which can be all found in the network. Thus, these references have not been listed in the manuscript.

5. Page 5/Line 17: It seems that for R2, the variability of winter haze day becomes larger in recent years as compared to the first and second periods.

**Reply:** It is actually true and it is apparent in Figure 2, which needs further studies in the future.

6. Page 6/Line 4: A little bit more discussion regarding the influence of Arctic sea ice loss on atmospheric circulation anomalies over east China is needed.

**Reply: [P6 L6-11]** Thanks for the suggestions. Actually, there are already some

discussions can be found in section 4 for this perspective. As the suggestion, more discussion regarding this has been added in the current version of this study as "Early studies (e.g., Wang et al., 2015) have indicated that the reduction of autumn ASI can lead to positive sea level pressure anomalies in mid-latitude Eurasia, northward shift of track of cyclone activity in China and weak Rossby wave activity in eastern China during winter season. These atmospheric circulation changes favor less cyclone activity and more stable atmosphere in eastern China, resulting in more haze days there."

Wang, H. J., Chen, H. P., and Liu, J.: Arctic sea ice decline intensified haze pollution in eastern China. Atmos. Oceanic Sci. Lett., 8, 1-9, 2015.

7.  Page 6/Line 14-15: I suggest the authors add some discussion about the possible factors contributing to increased precipitation during the first and second period and decreased precipitation during the third period.

**Reply:** Thanks for the suggestions. However, this article is mainly focused on the discussion about the influences of precipitation on the haze pollution, not the reasons for the precipitation changes. This suggestion as you proposed is beyond the scope of this article, thus the discussions about the precipitation changes have not been added in the current version.

8.  It is not clear how the authors define ">" and "<" signs in Figure 6? Relative to what? Please clarify.

**Reply: [P19]** (1) ">" means "larger than" and "<" means "less than" in Figure 6, which have been clarified in the figure caption.

(2) The comparisons are implemented among these three periods that discussed in the paper, i.e. the second period is compared with the first period and the third period is compared with the second period.

9.  Page 8/Line 15-21: I would like to suggest the authors to add more discussion

about the projected changes in precipitation and surface winds over east China in near term based on recent studies, CMIP5 model projections.

**Reply: [P8 L22-27]** Some discussions have been added as "Projections from CMIP5 models indicate that the low-level atmosphere tends to be more unstable and the atmosphere humidity will decrease in eastern China (Wang et al., 2015). Simultaneously, the winter precipitation in eastern China is projected to increase (Tian et al., 2015), but the surface winds decrease (Jiang et al., 2013). Thus there will be both favorable and unfavorable factors for haze occurrences in the near future based on the model projections. However, there is no doubt that, with the projected sea ice extent decrease (Kirtman et al., 2013), weakening of the winter East Asian monsoon wind (Wang et al., 2013) and total energy consumption increase, the haze pollution in eastern China may continue to be a serious problem in the near future. "

Wang, H. J., Chen, H. P., and Liu, J.: Arctic sea ice decline intensified haze pollution in eastern China. Atmos. Oceanic Sci. Lett., 8, 1-9, 2015.

Tian, D., Guo, Y., and Dong, W. J.: Future changes and uncertainties in temperature and precipitation over China based on CMIP5 models. Adv. Atmos. Sci., 32(4), 487-496, doi:10.1007/s00376-014-4102-7, 2015.

Jiang, Y., Luo, Y., and Zhao, Z. C.: Maximum wind speed changes over China. Acta Meteor. Sinica, 27(1), 63-74, doi:10.1007/s13351-013-0107-x, 2013.

Kirtman, B. et al.: Near-term Climate Change: Projections and Predictability. In: Climate Change 2013: The Physical Science Basis. Contribution of Working Group I to the Fifth Assessment Report of the Intergovernmental Panel on Climate Change [Stocker, T. F., Qin, D., Plattner, G.-K., Tignor, M., Allen, S. K., Boschung, J., Nauels, A., Xia, Y., Bex, V., and Midgley, P. M. (eds.)]. Cambridge University Press, Cambridge, United Kingdom and New York, NY, USA, pp. 953–1028, doi:10.1017/CBO9781107415324.023, 2013.

Wang H. J., He, S. P., and Liu, J. P.: Present and Future Relationship between the East Asian winter monsoon and ENSO: Results of CMIP5. J. Geophys. Res. Ocean, 118, 1-16, DOI:10.1002/jgrc.20332, 2013.

**Responses to Referee #2:**

The impact of climate change on air quality is an important cross-disciplinary topic. Previous studies were mainly conducted by using numerical models and there is limited number of studies based on analysis of observational data, especially in China. This study presented a very interested analysis on the trend of number of haze days and investigated the influences from Artic sea ice extent, precipitation and surface wind speed for different decades based on measurements at 756 ground station during 1960-2012. This paper is generally well-written and provides a different but unique angle of view from climatologist on the trend of air pollution in eastern China. I would like to recommend its publication in Atmospheric Chemistry and Physics after the following minor/technical points addressed appropriately.

**Reply:** Thanks for your suggestions, which have been addressed point by point in the following and the corresponding corrections have been presented in the manuscript.

Main comments:

1) If I understand it correctly, the "haze day data" (i.e. the term defined as "monthly haze day data" at L1, Page 4.) used in the trend analysis is the "total number of haze day in a month". In the text, this term is not very clear defined. It will be better to give some details of the data in Sect. 2 and clarified this term because it is not widely used in countries other than China. Also, it will be better to include some previous works e.g. Chen et al., 2015, which used the same dataset, as a reference in this section.

**Reply:** (1) Yes, the "monthly haze day" referred in this study actually is the "total number of haze in a month", which has been clarified in the paper.

(2) The previous works (e.g. Chen et al., 2015) as you referred is the paper by Chen and Wang, 2015, JGR? If yes, we should clarify that the haze day in that work is defined using visibility, relative humidity, and wind speed, which is different from the data that used in this study. However, similar results can be found between these two datasets. The haze days that used in this study are generally determined according to the immediately weather phenomenon by the sophisticated observers, which have been used in another

work (Wang et al., 2015). These corrections have been clarified in the current version of the manuscript.

2)  The discussions in the last paragraph of Sect. 3 (L3-L19, Page 7): The authors tried to discuss the reasons of the contradict trends between haze pollution and emission control. One point worthwhile to be mentioned here is that the trend of haze pollution based on "haze day data" actually is the trend of frequency (of haze day) but not the averaged pollution concentrations. The former might link more with the change in occurrence frequency of extremely stagnant weather, which was influenced by natural climate variability (Zhang et al., 2016), but the latter will be more related to the emission and control measures. Here a comparison with the two variables should consider the differences, at least mention the possible influence. In additional, this part probably can be moved into the Section 4 as the last discussion point highlighted for policy makers. Reference: Zhang, Y. et al., Impact of synoptic weather patterns and inter-decadal climate variability on air quality in the North China Plain during 1980-2013, Atmos. Environ.,124, 119-128, 2016.

**Reply:** As your suggestion, more discussions have been added and this part has been also moved to Section 4.

Minor comments:

1)  Line 21-22, page 2: "particulate matter 2.5", Please change it to "fine particulate matter (PM2.5)" or "particulate matter with mean aerodynanmic diameter less than 2.5 micrometers".

**Reply:** Corrected.

2)  Line 20-21, Page 3: "a recent study further reveals that", should you have a reference here? For example, "a recent study by Wang (2015) further revealed that".

**Reply:** Corrected.

3)  Figure 6: What does "Jiangnan (JN)" in the figure notes mean? Maybe change it to "South China".

**Reply:** Corrected. "Jiangnan (JN)" has been changed to R2 and "Huang-Huai (HH)" to

1        R1, which have been labeled in Figure 1.

[revised manuscript text omitted]